

# Large-scale comparative visualisation of sets of multidimensional data

Dany Vohl[1], David G. Barnes[2,3], Christopher J. Fluke[1,2], Govinda Poudel[4], Nellie Georgiou-Karistianis[4], Amr H. Hassan[1], Yuri Benovitski[2], Tsz Ho Wong[2], Owen L. Kaluza[2], Toan D. Nguyen[2] and C. Paul Bonnington[2]

[1] Centre for Astrophysics & Supercomputing, Swinburne University of Technology, Hawthorn, Victoria, Australia
[2] Monash eResearch Centre, Monash University, Clayton, Victoria, Australia
[3] Faculty of Information Technology, Monash University, Clayton, Victoria, Australia
[4] School of Psychological Sciences, Monash University, Clayton, Victoria, Australia

## ABSTRACT

We present *encube*—a qualitative, quantitative and comparative visualisation and analysis system, with application to high-resolution, immersive three-dimensional environments and desktop displays. *encube* extends previous comparative visualisation systems by considering: (1) the integration of comparative visualisation and analysis into a unified system; (2) the documentation of the discovery process; and (3) an approach that enables scientists to continue the research process once back at their desktop. Our solution enables tablets, smartphones or laptops to be used as interaction units for manipulating, organising, and querying data. We highlight the modularity of *encube*, allowing additional functionalities to be included as required. Additionally, our approach supports a high level of collaboration within the physical environment. We show how our implementation of *encube* operates in a large-scale, hybrid visualisation and supercomputing environment using the CAVE2 at Monash University, and on a local desktop, making it a versatile solution. We discuss how our approach can help accelerate the discovery rate in a variety of research scenarios.

## BACKGROUND

Scientific visualisation (*McCormick, 1988*; *Frenkel, 1988*; *DeFanti, Brown & McCormick, 1989*) is now well established as a means for gaining the required insight from multidimensional data. Visual insights can be obtained via different techniques. For a comprehensive review of the variety of standard techniques see, for example, *Akenine-Möller, Haines & Hoffman (2008)* for an overview of practices for real-time rendering; see *Toriwaki & Yoshida (2009)* for the mathematical theory of image processing and related algorithms; and *Szeliski (2010)* for common techniques about image analysis and interpretation. In this work, we distinguish between four broad classes of scientific visualisation. *Qualitative visualisation* allows an intuitive understanding of the data by displaying simple shapes and colours. *Quantitative visualisation* supports further comprehension of the data by connecting visual information with inherent measured quantities (e.g., using a colour bar, mean value of a selected region of interest, etc.).

Corresponding authors
Dany Vohl, dvohl@swin.edu.au
David G. Barnes,
david.g.barnes@monash.edu

*Comparative visualisation* emphasises the investigation of similarities or differences between two or more datasets. Finally, *collaborative visualisation* refers to situations where a group of researchers work together to organise, analyse, debate, and make sense of data collectively.

Structured three dimensional (3D) images or *data cubes* have an increasing presence in scientific research. Medical imaging (e.g., *Udupa & Herman, 1999*) routinely generates data cubes using instruments like positron emission tomography (PET), computerized tomography (CT) and magnetic resonance imaging (MRI). Ecology, oceanography and Earth sciences use hyperspectral remote sensing from airborne and satellite systems to gather data cubes formed of two spatial and one spectral dimensions (e.g., *Craig et al., 2006*; *Schalles, 2006*; *Govender, Chetty & Bulcock, 2007*; *Adam, Mutanga & Rugege, 2010*). Astronomers gather data cubes from instruments and facilities like integral field spectrographs (e.g., *Bacon et al., 2001*; *Bryant et al., 2012*), and radio interferometers (e.g., *Thompson, Moran & Swenson Jr, 2008*).

We use the term data cube *survey* when referring to scenarios where multiple data cubes are generated or collected. While the growth in size and quantity of data cubes in surveys allow novel and ambitious science to be undertaken, significant challenges are posed for knowledge discovery and analysis. As the number of data cubes increases, the classical desktop-based methodology—one data cube is examined at a time by a single person—is inefficient (*Isenberg et al., 2011*). Instead, researchers want to rapidly and interactively compare subsets of data cubes concurrently. To make sense of data in large survey projects, analysis and interpretation is rarely done alone: researchers want to explore and analyse data collaboratively, either synchronously or asynchronously, as well as discuss and debate their findings (*Heer & Agrawala, 2008*).

One of the primary goals of any data cube survey is to identify and investigate similarities and differences between individual participants, objects or observations. Additionally, it may be relevant to compare empirical or numerical models with the input data cubes. As instrumentation, facilities and data collection practices improve the number of data cubes per survey has grown rapidly. To give a sense of the magnitude of different surveys, let us focus on two representative examples from neuroscience and astronomy.

The IMAGE-HD study (*Georgiou-Karistianis et al., 2013*) is a multi-modal MRI longitudinal study of Huntington's disease (HD, *Walker, 2007*). The IMAGE-HD study investigates links between brain structure, microstructure and brain function with clinical, cognitive and motor deficits in both pre-symptomatic and symptomatic individuals with HD. Structural, functional, diffusion tensor, and susceptibility weighted MRI images have been acquired at three time points in over 100 volunteers at the start of the study, and after 18 and 30 months (e.g., *Domínguez D et al., 2013*; *Poudel et al., 2013*).

The Westerbork HI survey of spiral and irregular galaxies (WHISP, *Van der Hulst, Van Albada & Sancisi, 2001*; *Swaters et al., 2002*) utilised the Westerbork Synthesis Radio Telescope to study the neutral hydrogen (HI) content in ∼500 nearby galaxies. The goal of the survey was to investigate how the density distribution and motion of neutral hydrogen (HI) in galaxies depends on their shape (morphology), radio luminosity and environment (*Van der Hulst, Van Albada & Sancisi, 2001*). A typical WHISP spectral data cube comprised 512 × 512 spatial pixels and 128 frequency channels. With the upgrade to

the APERTIF instrument (*Verheijen et al., 2009*), the survey will grow to 20,000 data cubes with a two thousand-fold increase in voxels per data cube.

These are two of many projects that includes large multi-dimensional datasets. For more projects from medical imaging, see for example *Evans et al. (1993)*; *Evans (2006)*, and *Essen et al. (2012)*. Similarly, for more projects from astronomy, see for example *Booth et al. (2009)*, *Johnston et al. (2008)*, and *Quinn et al. (2015)*.

For comparative visualisation of large sets of data, high resolution displays (including tiled displays) have been introduced as an effective support (*Son et al., 2010*; *Lau et al., 2010*; *Fujiwara et al., 2011*; *Gjerlufsen et al., 2011*). However, previous research primarily focussed on how to render a visualisation seamlessly over multiple displays, and how to interact with the visualisation space. In the context of comparative visualisation of large sets of data using high resolution display environments, some questions remain open. How can comparative visualisation and analysis be integrated into a unified system? How can the discovery process be documented? Finally, how can we enable scientists to continue the research process once back at their desktop? In this work, we present a visualisation and analysis system that responds to all three questions.

## Motivation

The main motivation for our work was the identified need to accelerate visualisation led discovery and analysis of data cube surveys in two specific application areas. The first scenario was to enable new insights into the effect of HD on the microstructural integrity of the brain white matter, through large-scale interactive, comparative visualisation of Magnetic Resonance Imaging (MRI) data from IMAGE-HD. The second scenario was to support the first ever, large-scale systematic morphological (i.e., shape-based) classification of the kinematic structures of galaxies, commencing with the WHISP survey (V Kilborn, pers. comm., 2015–2016).

Although the data collection methods and interpretation are very different, both projects required a visualisation solution that:

1. provided a visual overview of the entire data cube survey, or a sufficiently large sub-set of the survey;
2. allowed qualitative, quantitative, and comparative visualisation;
3. supported interaction between the user and the data, including volume rotation, translation and zoom, modification of visualisation properties (colour map, transparency, etc.), and interactive querying;
4. supported different ways to organise the data, including automatic and manual organisation of data within the display space, as well as different ways of sorting lists using metadata included with the dataset;
5. could utilise stereoscopic displays to enhance comprehension of three-dimensional structures;
6. allowed a single data cube to be selected from the survey and visualised at higher-resolution;
7. encouraged collaborative investigation of data, so that a team of expert researchers could rapidly identify the relevant features;

8. was extensible (i.e., easily customizable) so that new functionalities can be easily be added as required;

9. automatically tracked the workflow, so that the sequence of interactions could be recorded and then replayed;

10. was sufficiently portable that a single solution could be deployed in different display environments, including on a standard desktop computer and monitor.

Identifying comparative visualisation of many data cubes (Item 1) as the key feature that could provide enhanced comprehension and increase the potential for collaborative discovery (Item 7), we elected to work with the CAVE2[TM][1] at Monash University in Melbourne, Australia (hereafter Monash CAVE2). To support quantitative interaction (Items 2, 3 and 4), we decided to work with a multi-touch controller to interact, query, and display extra quantitative information about the visualised data cubes. The CAVE2 provides multiple stereo-capable screens (Item 5). To visualise a single data cube at higher-resolution (Item 6), we elected to work with Omegalib *Febretti et al. (2014)*, and the recently added multi-head mode of S2PLOT (*Barnes et al., 2006*) (see 'Related work', 'Process-Render Units'). For the solution to be easily extensible (Item 8), we designed a visualisation and analysis framework, where each of its parts can be customized as required (see 'Overview of encube' and 'Implementation'). The framework incorporates a manager that can track the workflow (Item 9), so that each action can be reviewed, and system states can be reloaded. Finally, the framework can easily be scaled-down for standalone use with a standard desktop computer.

[1]CAVE2[TM] is a trademark of the University of Illinois Board of Trustees.

## The CAVE2

The CAVE2 is a hybrid 2D/3D virtual reality environment for immersive simulation and information analysis. It represents the evolution of immersive virtual reality environments like the CAVE (*Cruz-Neira et al., 1992*). A significant increase in the number of pixels and the display brightness is achieved by replacing the CAVE's use of multiple projectors with a cylindrical matrix of stereoscopic panels. The first CAVE2 Hybrid Reality Environment (*Febretti et al., 2013*) was installed at the University of Illinois at Chicago (UIC). The UIC CAVE2 was designed to support collaborative work, operating as a fully immersive space, a tiled display wall, or a hybrid of the two. The UIC CAVE2 offered more physical space than a traditional five or six wall CAVE, better contrast ratio, higher stereo resolution, more memory and more processing power.

The Monash CAVE2 is an 8-meter diameter, 320 degree panoramic cylindrical display system. It comprises 80 stereo-capable displays arranged in 20 four-panel columns. Each display is a Planar Matrix LX46L 3D LCD panel, with a $1,366 \times 768$ resolution and 46'' in diagonal. All 80 displays provide a total of $\sim$84 million pixels. For image generation, the Monash CAVE2 comprises a 20-node cluster, where each node includes dual 8-core CPUs, 192 gigabytes (GB) of random access memory (RAM), along with dual 1536-core NVIDIA K5000 graphics cards. Approximately 100 tera floating-point operations per second (TFLOP/s) of integrated graphic processing units (GPU)-based processing power is available for computation. Communication between nodes is through a 10 gigabit (Gb) network fabric. Data is stored on a local Dell server, containing 2.2 terabytes (TB) of

ultrafast redundant array of independent solid-state disk (RAID5 SSD), and 14 TB of fast SATA drives (RAID5). This server has 60 Gbit/s connectivity to the CAVE2 network fabric. It includes 256 GB of RAM to cache large files, and transfer out to the nodes very rapidly.

## Outline

The remainder of this paper is structured as follow. 'Related work' reviews related work and makes connections to our requirements. 'Overview of encube' presents a detailed description of the architecture design of encube, our visualisation and analysis system. 'Implementation' presents our implementation in the context of the Monash CAVE2. 'Timing experiment' presents a timing experiment comparing performances of encube when executed within the Monash CAVE2 and on a personal desktop. Finally, 'Discussion' discusses the proposed design and implementation, and discusses how it can help accelerate the discovery rate in a variety of research scenarios.

## RELATED WORK

We focus our attention on literature related to comparative visualisation, the use of high resolution displays (including tiled displays and CAVE2-style configurations), and interaction devices.

Early work by *Post & Van Wijk (1994)*, *Hesselink, Post & Van Wijk (1994)*, and *Pagendarm & Post (1995)* discussed comparative visualisation using computers, particularly in the context of tensor fields and fluid dynamics. They discussed the need to develop comparative visualisation techniques to generate comparable images from different sources. They found that two approaches to comparative visualisation can be considered: image-level comparison, visually comparing either two observation images or a theoretical model and an observation image, all generated via their own pipeline; and data level comparison, where data from two different sources are transformed to a common visual representation via the same pipeline.

*Roberts (2000)* and *Lunzer & Hornbæk (2003)* advocated the use of side-by-side interactive visualisations for comparative work. *Roberts (2000)* proposed that multiple views and multiform visualisation can help during data exploration—providing alternative viewpoints and comparison of images, and encouraging collaboration. *Lunzer & Hornbæk (2003)* identified three kinds of issues common to comparative visualisation during exploration of information: a high number of required interactions, difficulty in remembering what has been previously seen, and difficulty in organising the exploration process. We return to these issues in 'Discussion' with regards to our solution. In accordance with *Roberts (2000)*, they argued that these issues can be reduced through an interactive interface that lets the user modify and compare different visualisations side by side. *Unger et al. (2009)* put these principles to work by using side-by-side visualisation to emphasize the spatial context of heterogeneous, multivariate, and time dependent data that arises from a spatial simulation algorithm of cell biological processes. They found that side-by-side views and an interactive user interface empowers the user with the ability to explore the data and adapt the visualization to his current analysis goals.

For scientific visualisation, *Ni et al. (2006)* highlighted the potential of large high-resolution displays as they offer a way to view large amounts of data simultaneously with the increased number of available pixels. They also mention that benefits of large-format displays for collaborative work—as a medium for presenting, capturing, and exchanging ideas—have been demonstrated in several projects (e.g., *Elrod et al., 1992*; *Raskar et al., 1998*; *Izadi et al., 2003*). Moreover, *Chung et al. (2015)* assess that the use of multiple displays can enhance visual analysis through its discretized display space afforded by different screens.

In this line of thinking, *Jeong et al. (2005)*, *Doerr & Kuester (2011)*, *Johnson et al. (2012)*, *Marrinan et al. (2014)* and *Kukimoto et al. (2014)* discussed software for cluster-driven tiled-displays called SAGE, CGLX, DisplayCluster, SAGE2, and HyperInfo respectively. These solutions provide dynamic desktop-like windowing for media viewing. This includes media content such as ultra high-resolution imagery, video, and applications from remote sources to be displayed. They highlight how this type of setting lets collective work to take place, even remotely. *Febretti et al. (2014)* presented Omegalib to execute multiple immersive applications concurrently on a cluster-controlled display system, and have different input sources dynamically routed to applications.

In the context of comparative visualisation using a cluster-driven tiled-display, *Son et al. (2010)* discussed the visualisation of multiple views of a brain in both 2D and 3D using a 4 × 2 tiled-display, with individual panel resolution of 2,560 × 1,600 pixels. Their system supported both high resolution display over multiple screens (using the pixel distribution concept from SAGE and the OpenGL context distribution from CGLX) and multiple image display. Supporting both a 3D mesh and 2D images, they provided region selection of an image, which highlighted the same region in other visualisations of the same object. They also included mechanisms to control of the 3D mesh (resolution, position and rotation), and control of image scale using a mouse and keyboard, enabling users to modify the size of images displayed. Other approaches are presented by *Lau et al. (2010)*, *Fujiwara et al. (2011)*, and *Gjerlufsen et al. (2011)*, who developed applications to control multiple visualisations simultaneously.

*Lau et al. (2010)* used ViewDock TDW to control multiple instances of the Chimera software (http://www.cgl.ucsf.edu/chimera) to visualise dozens of ligand–protein complexes simultaneously, while preserving the functionalities of Chimera. ViewDock TDW permitted comparison, grouping, analysis and manipulation of multiple candidates—which they argued increases the efficacy and decreases the time involved in drug discovery. In an interaction study within a multisurface interactive environment called the WILD room, *Gjerlufsen et al. (2011)* discussed the comparative visualisation of 64 brain models. Controlling the Anatomist (http://brainvisa.info/web/index.html) software with the SubstanceGrise middleware, they enabled synchronous interaction with multiple displays using several interaction devices: a multitouch table, mobile devices, and tracked objects. Discussing these results, *Beaudouin-Lafon (2011)* and *Beaudouin-Lafon et al. (2012)* note that large display surfaces let researchers organize large sets of information and provide a physical space to organize group work.

Similarly, *Fujiwara et al. (2011)* discuss the development of a multi-application controller for the SAGE system, with synchronous interaction of multiple visualisations. A remote application running on personal computer was used to control the *SAGE* system. They evaluated how effectively scientists find similarities and differences between visualised cubic lattices (3D graph with vertices connected together with edges) using their controller. They measured the time required to complete a comparison task in two cases where the controller controls: (1) all visualisations synchronously; or (2) only one visualisation at a time. For their test, they built a tiled display of $2 \times 2$ LCD monitors and three computers. For both cases, participants would look at either 8 or 24 volumes displayed side-by-side, and spread evenly on the different displays. Their results showed that comparison using synchronized visualisation was generally five times faster than individually controlled visualisation (case 1: ∼25 seconds per task; case 2: ∼130 seconds per task).

Figure 1 depicts system design schematics from *Son et al. (2010)*, *Lau et al. (2010)*, *Fujiwara et al. (2011)* and *Gjerlufsen et al. (2011)*. All designs depend upon a one-way communication model, where a controller sends a command to control how visualisations are rendered on the tiled-display. Hence, all models primarily focussed on enabling interaction with the visualisation space. With these models, the controller cannot receive information back from the display cluster (or equivalent computing infrastructure). It is worth noting that designs from *Son et al. (2010)*, *Fujiwara et al. (2011)* and *Gjerlufsen et al. (2011)* also enable a visualisation to be rendered over multiple displays.

To interact with visualisations, many interaction devices have been proposed, varying with the nature of the visualisation environment and the visualisation space (e.g., 2D, 3D). One of the most common method is based on tracked devices (e.g., tracked wand controllers, gloves and body tracking, smart phone) that allow users to use natural gestures to interact with the visual environment (e.g., *LaViola et al., 2004*; *Roberts et al., 2010*; *Nancel et al., 2011*; *Febretti et al., 2013*). For instance, both the CAVE2 and the WILD room use a tracking system based on an array of Vicon Bonita infrared cameras to track the physical position of controllers relative to the screens. To interact with 3D data, another interaction method involves sphere shaped controllers that enables 6 degrees of freedom (e.g., *Amatriain et al., 2009*; *Febretti et al., 2013*). However, *Hoang, Hegie & Harris Jr (2010)* argued that the practicality of these methods diminishes as the set of user controllable parameters and options increase—and that precise interaction can be difficult due to the lack of haptic feedback.

Another interaction method involves the use of mobile devices such as phones and tablets with multitouch capabilities (e.g., *Roberts et al., 2012*; *Roberts, Wakefield & Wright, 2013*; *Donghi, 2013*). They are used to interact with the visual environment or to access extra information not directly visible on the main display. For example, *Höllerer, Kuchera-Morin & Amatriain (2007)* and *Amatriain et al. (2009)* provided concurrent users with configurable interfaces. It let users adopt distinct responsibilities such as spatial navigation and agent control. *Cheng, Li & Müller-Tomfelde (2012)* focused on an interaction technique that supports the use of tablet devices for interaction and collaboration with large displays. They presented a management and navigation interface based on an interactive world-in-miniature view and multitouch gestures. By doing so, users were able to manage their views

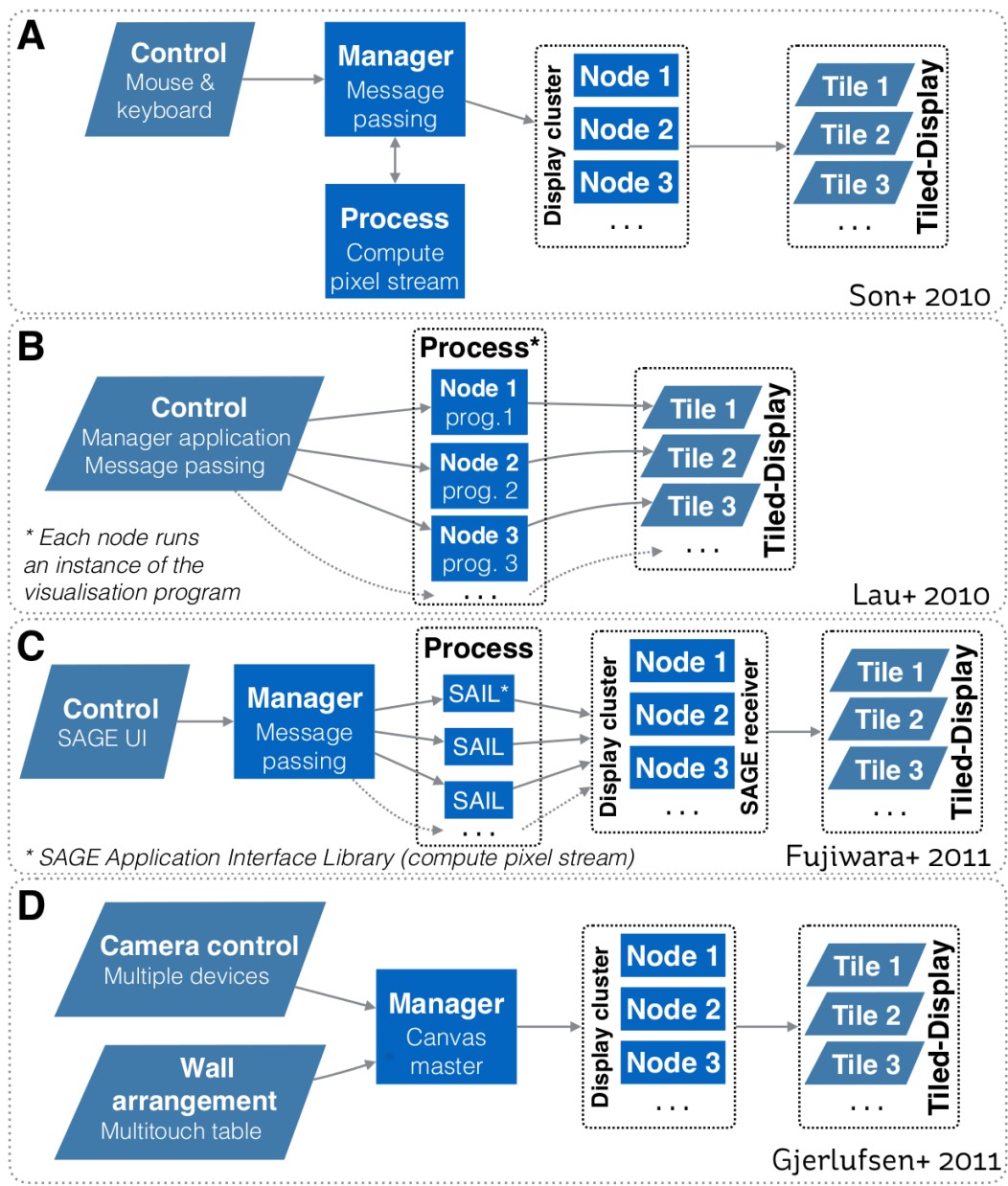

**Figure 1** Systems design schematics from: **(A)** *Son et al. (2010)*; **(B)** *Lau et al. (2010)*; **(C)** *Fujiwara et al. (2011)*; **and (D)** *Gjerlufsen et al. (2011)*. All designs are predicated upon a one-way communication model, where a controller sends a command to control how visualisations are rendered on the tiled-display. All models primarily focussed on enabling interaction with the visualisation space. Designs A, C and D also enable a visualisation to be rendered over multiple displays.

on their tablets, navigate between different areas of the workspace, and share their view with other users. As multitouch devices are inherently 2D, a limitation of models based on mobile devices—when working with 3D data—is related to precision in manipulation and region selection.

From this review, we can note that previous research primarily focussed on *rendering a visualisation seamlessly over multiple displays*, and in cases involving comparative visualisation, it placed an emphasis on *enabling interaction with the visualisation space*. As our focus is to accelerate visualisation led discovery and analysis of data cube surveys, it is also important to consider the following questions. *How can we integrate comparative visualisation and analysis into a unified system? How can we document the discovery process?* Finally, *how can we enable scientists to continue the research process once back at their desktop*? We cover our proposed solutions to these three question throughout the following sections.

## OVERVIEW OF ENCUBE

encube's design focuses on enabling interactivity between the user and a date cube survey, presented in a comparative visualisation mode within an advanced display environment. Its main aim is to accelerate the visualisation and analysis of data cube surveys –with an initial focus on applications in neuroimaging and radio astronomy. The system's design is modular, allowing new features to be added as required. It comprises strategies for qualitative, quantitative, and comparative visualisation, including different mechanisms to organise and query data interactively. It also includes strategies to serialize the workflow, so that actions taken throughout the discovery process can be reviewed either within the advanced visualisation environment or back at the researcher's desk. We discuss the general features of encube's design before addressing implementation in 'Implementation.'

We abstract the system's architecture into two layers: a *process layer*, and an *input/output layer* (Fig. 2). This abstraction enables tasks to be executed on the most relevant unit based on the nature of the task (e.g., compute intensive, communication). While sharing similarities with other systems (e.g., *Jeong et al., 2005*; *Fujiwara et al., 2011*; *Gjerlufsen et al., 2011*; *Febretti et al., 2014*; *Marrinan et al., 2014*), it differs through three main additions (Table 1):

1. mechanisms for quantitative visualisation;
2. mechanisms for comparative visualisation; and
3. mechanisms to serialize the workflow.

### Process layer

As shown in Fig. 2, the Process layer is the central component of our design. Communication occurs between a *Manager Unit* and one or more *Process-Render Units*. It follows a master/slaves communication pattern, where the *Process-Render Units* act upon directives from the *Manager Unit*.

*Process-Render Unit.* This component is the main computation engine. Each *Process-Render Unit* has access to data and is responsible for rendering to one or more *Display*

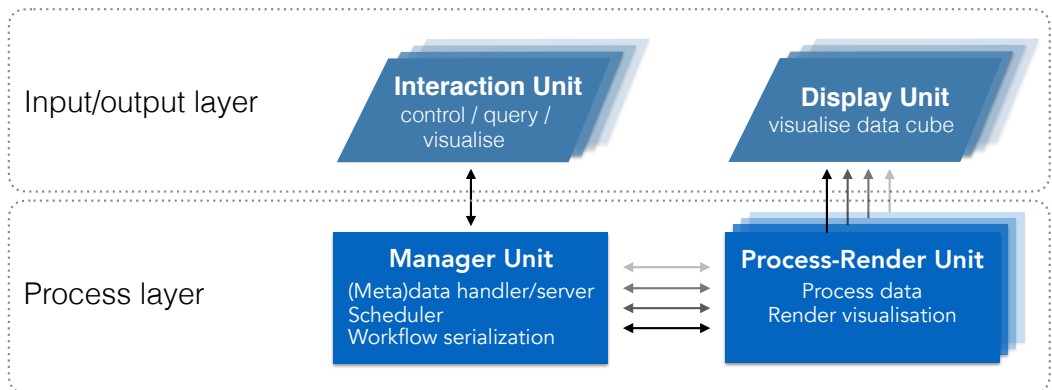

**Figure 2  System design schematic.** Our solution integrates visualisation and analysis into a unified system. The input/output layer includes units to visualise data (Display Units) and interact with visualisations (Interaction Units). The Process layer includes units to manage communication and data (Manager Unit), and to process and render visualisations (Process-Render Units). This system has two-way communication, enabling the *Interaction Unit* to query both the *Manager Unit* and the *Process-Render Units*, and retrieve the requested information. This information can then be reported on the *Interaction Unit* directly, reducing the amount of information to be displayed on the *Display Units*.

*Units.* As it is expected that most of the available processing power of the system will reside on the *Process-Render Units*, derived data products like a histogram, or quantitative information (e.g., mean voxel value) are also computed within this part of the Process layer.

*Manager Unit.* This component has three different functionalities. Foremost, it is the metadata server for the system, where a structured list of the dataset is available and can be served to *Process-Render Units* and *Interaction Units*. Secondly, it acts as a scheduler, where it manages and synchronizes tasks communication between *Process-Render Units* and *Interaction Units*. Tasks are handled following the queue model. Finally, as all communication and queries (e.g., visualisation parameter change) circulate via the *Manager Unit*, the workflow serialization occurs here.

## Input/output layer

*Interaction Units.* This component is in charge of controlling what to display, and how to display it. It also provides mechanisms to query the distributed data. In this context, screen-based controllers—laptops, tablets and smart phones—can both control the visualised content, and display information derived or computed from this content. The *Interaction Unit* can request that a specific computation be executed on one or more *Process-Render Units* and the result returned to the Interaction Unit. This result can then be visualised independently of the rendered images on the *Display Units*. While a single *Interaction unit* is likely to be the prefered option, multiple clients should be able to interact with the system concurrently.

*Display Units.* This component is where the rendered images and meta-data such as a data cube identification number (ID) are displayed. A *Display Unit* can display a single or

Vohl et al.
2016
10.7717/peerj-cs.88

**Table 1  Comparing systems from *Son et al. (2010)*, *Lau et al. (2010)*, *Fujiwara et al. (2011)*, *Gjerlufsen et al. (2011)*, and this work based on our user requirements.** Acronyms: query one volume (QOV); query multiple volume (QMV); modify visualisation property (e.g., colormap, transparency; MVP). A blank indicates that a topic is not discussed in related literature. Reorder is a cascading reordering event after manually moving a volume from one screen to another (see 'Manual data reordering').

| Requirement | | Son+ | Lau+ | Fujiwara+ | Gerjlufsen+ | This work |
|---|---|---|---|---|---|---|
| Visualisation class | Qualitative | • | • | • | • | • |
| | Quantitative | | | | | • |
| | Comparative | • | • | • | • | • |
| Interaction method (with volumes) | Rotate | • | • | • | • | • |
| | Pan/zoom | • | • | • | • | • |
| | MVP | | • | | | • |
| | QOV | | | | | • |
| | QMV | | | | | • |
| Organise visualisations | Automated | • | • | • | • | • |
| | Manual | | | | • | • |
| Organisation mechanism (comparative vis.) | Swap | | | | • | • |
| | Sort | | • | | | • |
| | Reorder | | | | | • |
| Organise data | Automatic | | • | | | • |
| | Manual | | • | | | • |
| Stereoscopic screens | | | | | | • |
| Number of screens | | 4 × 2 | 5 × 4 | 2 × 2 | 8 × 4 | 20 × 4 |
| Multi-screens visualisation | | • | | • | • | • |
| Collaborative environment | | • | • | • | • | • |
| Customizable | | | | • | • | • |
| Workflow history | | | | | | • |
| Standalone (single desktop) | | | | | | • |

multiple data cubes. It can also display part of a data cube in cases where a data cube is displayed over multiple *Display Units*. In a tiled-display setting, a display unit comprises one or more physical screens. Several types of displays can be envisioned (e.g., high definition or ultra-high resolution screens; 2D or stereoscopic capable screens; multi-touch screens). For a traditional desktop configuration, the Display Unit may be a single application window on a desktop monitor.

## IMPLEMENTATION

In this section we describe and discuss our implementation of encube for use in the Monash CAVE2. In this context, our design is being mapped to the CAVE2's main hardware components (Table 2). The Process layer is represented by a server node (*Manager Unit*) communicating with a 20-node cluster (*Process-Render Units*). Each node is controlling a column comprising four 3D-capable screens (*Display Units*). Finally, laptops, tablets, or smart phones (*Interaction Units*) connect to the *Manager Unit* using the Hypertext Transfer Protocol (HTTP) via a web browser. On the software side, three main programs are connected together to form a visualisation and analysis application system: a data

**Table 2** encube's layers and units mapped to the Monash CAVE2 hardware.

| Layer | Unit | Monash CAVE2 hardware |
|---|---|---|
| Process layer | Process-Render Units | **20-node cluster**, where each node comprises: |
| | | Dual 8-core CPUs |
| | | 192 GB of RAM |
| | | 2304-core NVIDIA Quadro K5200 graphics card |
| | | 1536-core NVIDIA K5000 graphics cards |
| | Manager Unit | **Head node**, comprising: |
| | | Dual 8-core CPUs |
| | | 256 GB ram |
| | | 2304-core NVIDIA Quadro K5200 graphics card |
| Input/output layer | Interaction Units | **Tablets, smartphones, or laptops**, in particular: |
| | | iPad2 A1395 |
| | Display Units | **20 columns × 4 rows of stereoscopic screens**, |
| | | where each screen is: |
| | | Planar Matrix LX46L 3D LCD panel |
| | | 1,366 × 768 pixels |
| | | 46" diagonal |

Monash CAVE2's 80 screens

Process-Render Units

Interaction Units

Manager Unit

**Figure 3 Components of encube depicted in the context of the Monash CAVE2.** The *Interaction Units* permit researchers to control and interact with what is visualised on the 80 screens of the CAVE2. The *Interaction Units* communicate with the *Manager Unit* via HTTP methods. A command sent from a client is passed by the *Manager Unit* to the *Process-Render Units*, which execute the action. The result is either drawn on the *Display Units* or returned to the *Interaction Unit*.

processing and rendering program (incorporating volume, isosurface and streamline shaders –instantiated on every *Process-Render Unit*), a manager program (instantiated on the *Manager Unit*), and an interaction program (instantiated on the *Interaction Units*). Schematics of the implementation, and how it relates to the general design, are depicted in Figs. 3 and 4 respectively.

## Process-Render Units

All processing and rendering of volumetric data is implemented as a custom 3D visualisation application—encube-PR. As processing and rendering is compute intensive, encube-PR is

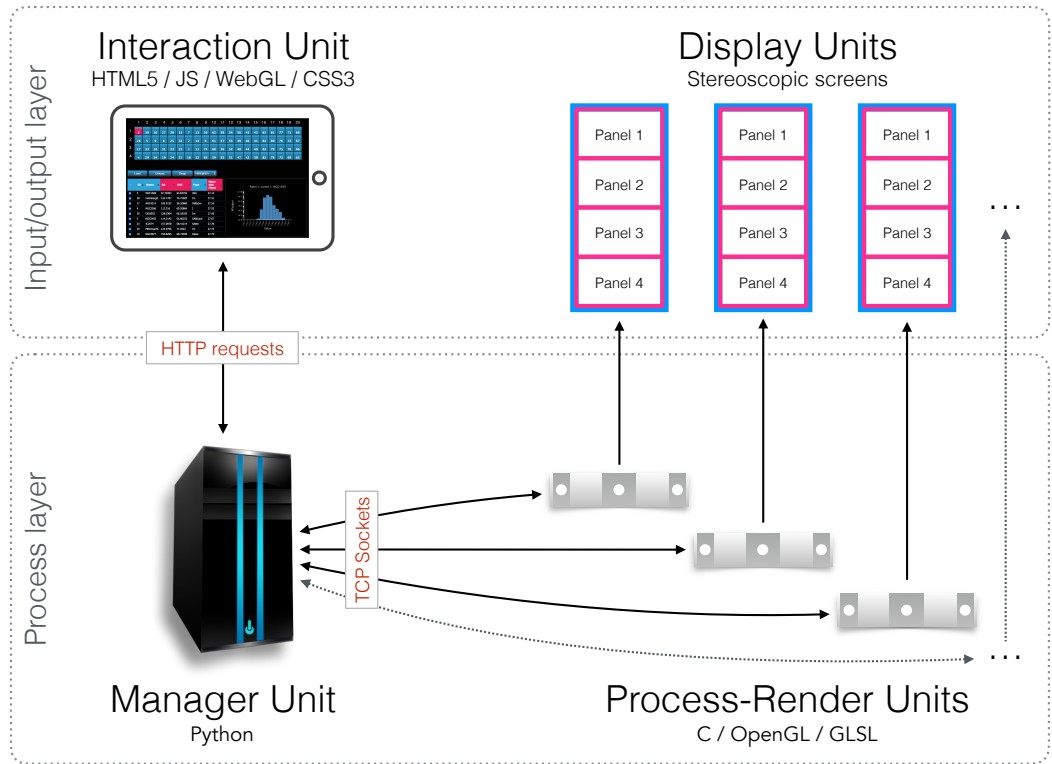

**Figure 4 Schematic diagram of the implementation.** An *Interaction Unit* is implemented as a web interface using HTML5, JavaScript, WebGL, and CSS3—it communicates with the *Manager Unit* through HTTP requests. The *Manager Unit* is a Python application—it communicates with *Process-Render Units* through TCP sockets. Each *Process-Render Unit* is implemented as a custom 3D visualisation application written in C, OpenGL and GLSL. The *Display Units* correspond to the 20 Planar Matrix LX46L 3D LCD panels of the Monash CAVE2.

written in C, OpenGL and GLSL, and based on S2PLOT (*Barnes et al., 2006*). Each Processor-Renderer node runs one instance of this application. The motivation behind our choice of using S2PLOT, an open source three-dimensional plotting library, is related to its support of multiple panels "out of the box" and stereoscopic rendering capabilities. It is also motivated by S2PLOT's multiple customizable methods to handle OpenGL (https://www.opengl.org) callbacks for interaction, and its support of remote input via a built-in socket. Moreover, S2PLOT can be programmed to share its basic rendering transformations (camera position etc.) with other instances. Custom shaders are supported in the S2PLOT pipeline, alongside predefined graphics primitives. Additionally, multi-head S2PLOT—a version of S2PLOT distributed over multiple *Process-Render Units*—or a binding with Omegalib (*Febretti et al., 2014*) enable the rendered visualisation to be distributed over multiple computer nodes and displays. The multi-head S2PLOT mode—developed specifically for encube—requires a configuration file with physical screen locations to render across mulitible S2PLOT instances via Message Passing Interface (MPI).

encube-PR is responsible for displaying dynamic, interactive 3D geometry such as isosurface and ray-casting volume rendering onto the 3D screens of the CAVE2. As one

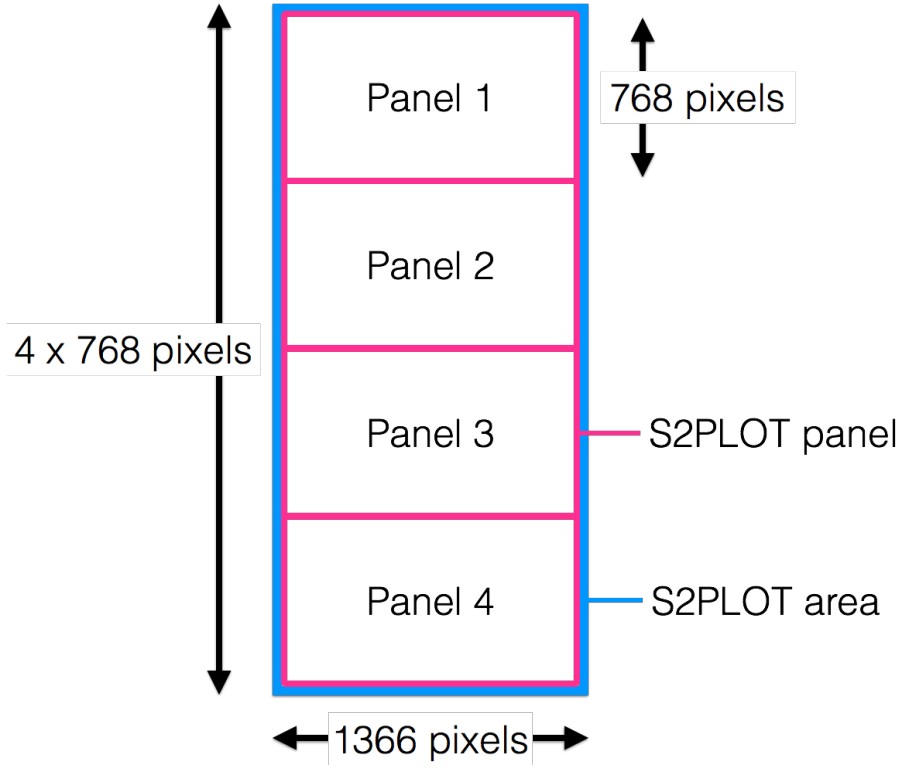

**Figure 5** **Drawing window covering all four panels.** S2PLOT area (blue) covering all four Planar Matrix LX46L 3D LCD panels of a column. Each screen has resolution of 1,366 × 768 pixels. The S2PLOT area is divided into four independent panels, with the size matched to the physical display resolution (pink).

instance of the application is executed per node (which controls four screens in the context of the Monash CAVE2), the plotting application generates one drawing window covering all four panels, and uses native S2PLOT functions to divide the entire drawing surface (device) into four panels vertically, matching the borders of the physical display device (i.e., the bezels of the screens). Hence, each S2PLOT panel is displayed on an individual screen in the CAVE2 display surface (Fig. 5).

Isosurface and ray-casting volume renderings are computed through custom GLSL fragments and vertex shader programs. The visualisation parameters in these programs can be modified interactively to highlight features of interest within the data (see 'Interaction Units'). Other custom shaders can be added to the volume rendering in order to overlay other types of data (e.g., flows or connective structures as streamlines in medical imaging—deployed for the IMAGE-HD element of this work; multi-wavelength 2D imaging data or celestial coordinate systems in astronomy).

## Manager Unit

The server, communication hub and workflow serialization is done in the *Manager Unit* application—encube-M. This application is implemented in Python (https://www.python.org) due to its ease for development and prototyping, along with its ability to execute C code when required. Python is also the principal control language for Omegalib

applications in the CAVE2; hence, encube-M offers the possibility for simple communication with Omegalib applications. Common multiprocessing (threading) and communication libraries such as Transmission Control Protocol (TCP) sockets or MPI for high-performance computing are also available. Furthermore, with its quick development and adoption by the scientific community in recent years (e.g., *Bergstra et al., 2010*; *Gorgolewski et al., 2011*; *Astropy Collaboration et al., 2013*; *Momcheva & Tollerud, 2015*), our system can easily integrate packages for scientific computing such as SciPy (e.g., NumPy, Pandas; *Jones, Oliphant & Peterson, 2001*), PyCUDA and PyOpenCL (*Klöckner et al., 2012*), semi-structured data handling (e.g., XML, JSON, YAML), along with specialized packages (e.g., machine learning for neuroimaging and astronomy; *Pedregosa et al., 2012*; *VanderPlas et al., 2012*; *Abraham et al., 2014*).

Throughout a session in the CAVE2, encube-M keeps track of the many components' states, and synchronizes the *Interaction Units* and the *Process-Render Units*. The *Manager Unit* communicates with the different *Process-Render Units* via TCP sockets, and responds to the *Interaction Units* via HTTP methods (request and response).

### *Workflow serialization*

Each step of a session is kept and stored to document the discovery process. This workflow serialization is done by adding each uniquely identified action (e.g., load data, rotate, parameter change) to a dictionary (key-value data structure) from which the global state can be deduced. The dictionary of a session can be stored to disk as a serialized generic Python object or as JSON data. Structured data can then be used to query and retrieve previous sessions, which can then be loaded and re-examined. This data enables researchers to review the sequence of steps taken throughout a session. It also makes it possible for researchers to continue the discovery process when they leave the large-scale visualisation environment and return to their desk.

## Interaction Units

We implemented the *Interaction Unit* as a web interface. Recent advances in web development technologies such as HTML5 (http://www.w3.org/TR/html5/), CSS3 (http://www.w3.org/TR/2001/WD-css3-roadmap-20010523/) and ECMAScript (http://www.ecma-international.org/memento/TC39.htm) (JavaScript) along with WebGL (https://www.khronos.org/webgl/) (a Javascript library binding of the OpenGL ES 2.0 API) enable the development of interfaces that are portable to most available mobile devices. This development pathway has the advantage of allowing development independent from the evolution of the mobile platforms ["write once, run everywhere" (*Schaaff & Jagade, 2015*)]. Furthermore, a web server can easily communicate with multiple web clients, enabling collaborative interactions with the system. As it is currently implemented, multiple *Interaction Units* can connect and interact with the system concurrently. However, we did not yet implement a more complex scheduler to make sure all actions are consistent with one another (future work).

In order to control the multiple *Process-Render Units*, we equipped the interface with core features essential for *interacting with 3D scenes* (rotate, zoom and translate a volume),

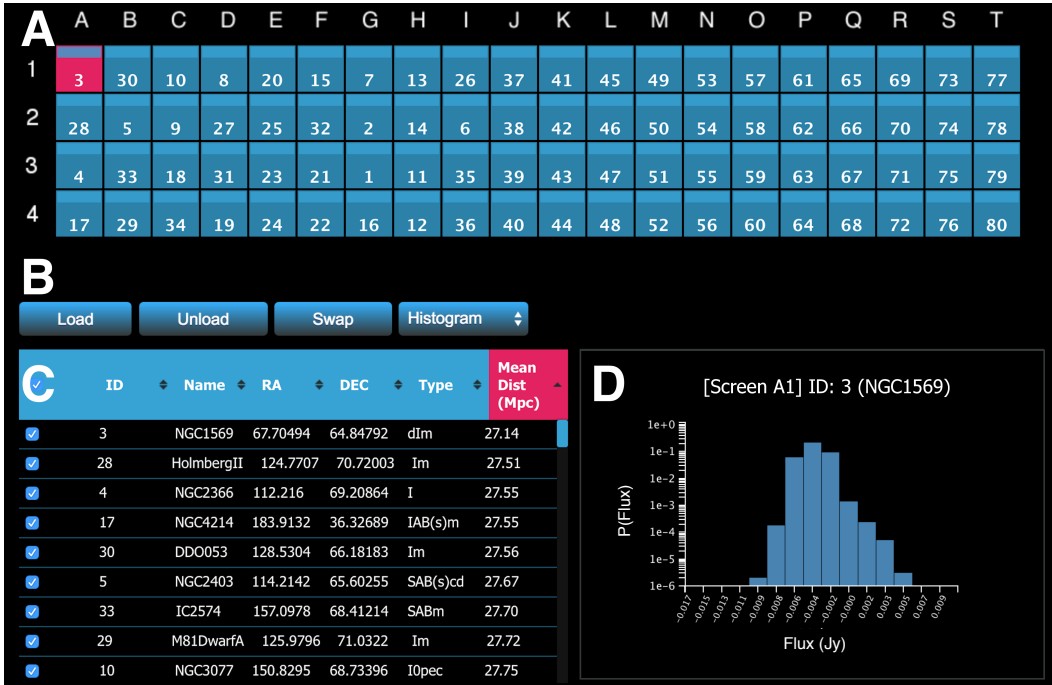

**Figure 6** **A screenshot of the meta-controller, configured for comparative visualisation of data from the WHISP (*Van der Hulst, Van Albada & Sancisi, 2001*), HIPASS (*Barnes et al., 2001*), and THINGS (*Walter et al., 2008*) galaxy surveys.** (A) A miniature representation of the Monash CAVE2's 4 rows by 20 columns configuration; (B) action buttons; (C) the dataset viewer (allows the survey to be sorted by multiple criteria); and (D) request/display quantitative information about data from one or multiple screens (e.g., a histogram is shown here for galaxy NGC1569 currently on screen A1).

*modifying the visualisation* (change colours, intensity, transparency, etc.), and *organising and querying* one or more data cubes. The interface comprises three main panels: a *meta-controller*, an *scene controller*, and a *rendering parameters controller*. We further describe the components and functionalities of the three panels in the next sections.

### Interactive data management: the meta-controller

The meta-controller panel enables the user to interactively handle data. Figure 6 shows an example interface of the meta-controller configured for morphological classification of galaxies. The panel comprises four main components.

The first component is a miniature representation of the CAVE2 (Fig. 6A). This component is implemented using Javascript and CSS3. Each display is mapped as a (coloured) rectangle onto a grid, representing the physical setting of the CAVE2 display environment (e.g., 4 rows by 20 columns). In the following, we refer to an element of this grid as a screen, as it maps to a screen location in the CAVE2. Indices for columns (A–T) and rows (1–4) are displayed around the grid as references.

The second component comprises action buttons (Fig. 6B). Actions include load data, unload data, swap screens, and query *Process-Render Units*. The action linked to each button will be described in the next section. The third component is an interactive table displaying the metadata of the data cube survey (Fig. 6C). Each metadata category is displayed as a

separate column. The table enables client-side multi-criteria sorting by clicking or tapping on the column header. In the example in Fig. 6, the data is simply sorted by the Mean Distance parameter in ascending order.

Finally, it is possible to query the *Process-Render Units* to obtain derived data products or quantitative information about the displayed data (Fig. 6D). The information returned in such a context is displayed in the lower right portion of the meta-controller module. The plots are dynamically generated using information sent back in JSON format from the web server and then drawn in a HTML canvas using Javascript.

### Interacting with the meta-controller

A user can manipulate data using several functionalities, namely load data to a screen, select a screen, unload data, swap data between screens, request derived data products or quantitative information, and manual data reordering.

*Load data.* Loading data onto the screens can be done via two methods:
1. Objects are selected from the table (Fig. 6C) and loaded by clicking or tapping the *load* button (Fig. 6B). This will display the selected objects in the order that they appear in the table (keeping track of the current sort state) in column-first, top-down order on the screens. The corresponding data cubes will then be rendered on the *Display Units* (Fig. 7). Figure 7A shows a picture of MRI imaging and tractography data from the IMAGE-HD study displayed on the CAVE2 screens, while Fig. 7C shows several points of view for different galaxy data taken from the THINGS galaxy survey (*Walter et al., 2008*), a higher resolution survey than the WHISP survey mentioned previously, but with only nearby galaxies (distance < 15 megaparsec[2]).
2. Load a specific cube via a click-and-drag gesture from the table and releasing on a given screen of the grid.
   Once data is loaded, each CAVE2 screen displays the ID of the relevant data cube.

[2]A parsec is distance measure equal to ~3.26 light years, or ~$3.086 \times 10^{13}$ km.

*Screen selection and related functionalities.* Each screen is selectable and interactive (such as the top-left screen in Fig. 6A). Selection can be applied to one or many screens at a time by clicking its centre (and dragging for multi-selection). Screen selection currently enables four functionalities:
1. *Highlight a screen within the CAVE2.* Once selected, a coloured frame is displayed to give a quick visual reference (Fig. 7B).
2. *Unload data.* By selecting one or more screens and clicking the *unload* button, the displayed data will be unloaded both within the controller and on the *Process-Render Unit*.
3. *Swap* (for comparative visualisation work). By selecting any two screens within the grid and clicking the *swap* button, the content (or absence of content if nothing is loaded on one of them) of the two displays will be swapped.
4. *Request derived data products or quantitative information* about a selected screen.

*Integrating comparative visualisation and analysis as a unified system.* As a proof of concept for the two-way communication, we currently have implemented two functionalities. The

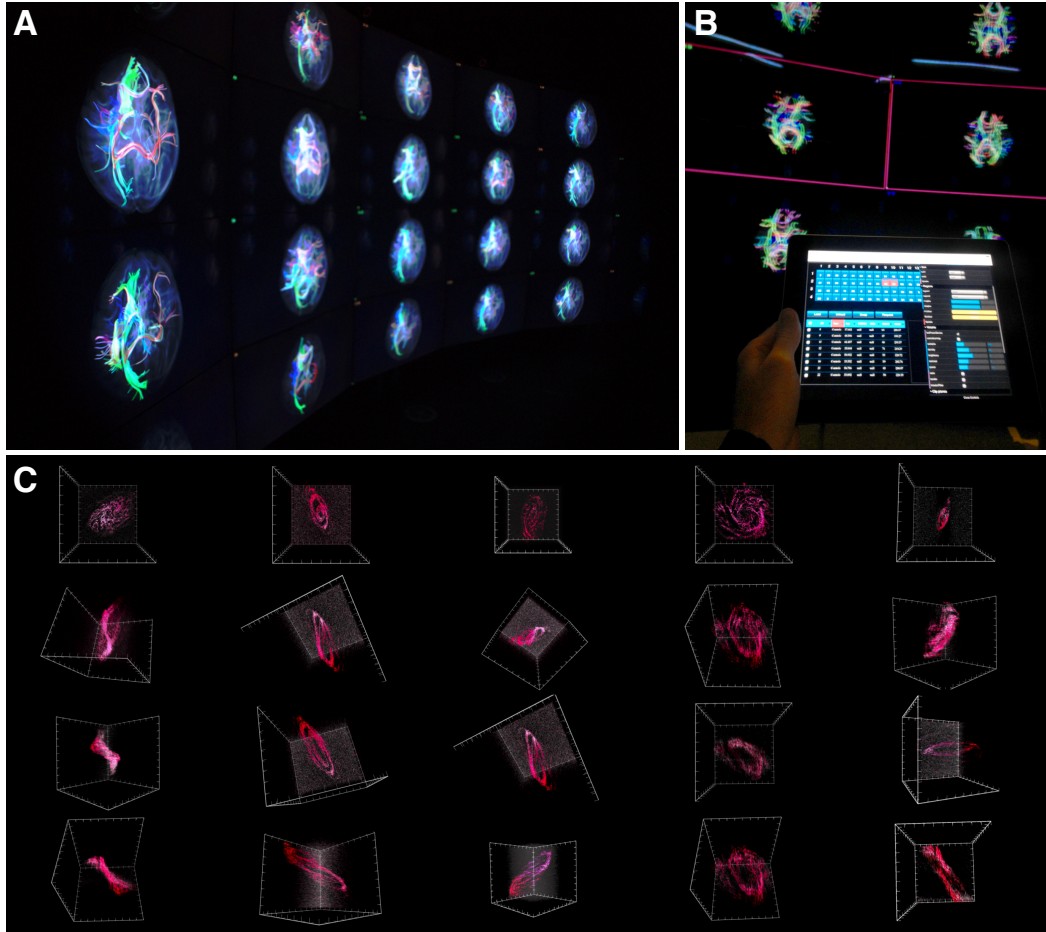

**Figure 7  encube in action.** (A) Photograph of a subset of five out of the 20 four-panel columns of the Monash CAVE2. (B) Selecting screens within the meta-controller leads to the display of a frame (pink) around the selected screens in the CAVE. (C) Visualisation outputs showing different galaxy morphologies taken from the THINGS galaxy survey (*Walter et al., 2008*).

first function queries a specific data cube to retrieve an interactive histogram of voxel values (Fig. 6D). The interactive histogram allows the user to dynamically alter the range of voxel values to be displayed. This approach is commonly used by astronomers working with low signal-to-noise data. The second function queries all plotted data cubes and summarizes the results in an interactive scatter plot (Fig. 8; see 'An example of execution' for further discussion). This example function enables neuroscientists to quickly access the number of connections between two given regions of the brain when comparing multiple MRI data files from control, pre-symptomatic and symptomatic individuals. Additional derived data products and quantitative information functionalities can easily be implemented based on specific user and use-case requirements.

*Manual data reordering.* To further support interactivity between the user and the environment, we included the option to manually arrange the order of the displayed data. For example, instead of relying only on automated sorting methods, a user may
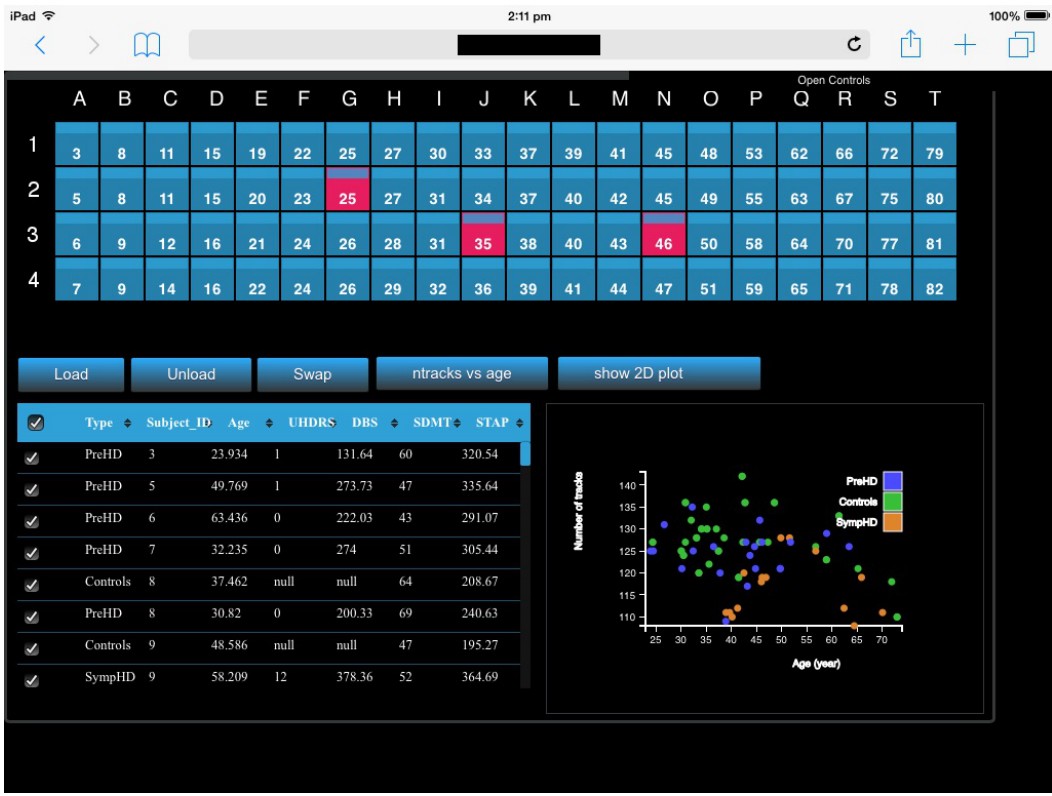

**Figure 8** **A screenshot of the iPad within the Monash CAVE2 showing the integration between analysis and comparative visualisation: querying all plotted brain data and summarizing the results in an interactive scatter plot.** In this example, neuroscientists can quickly summarize the number of plotted tracks per brain for 80 different individuals in relation to their age of the individual. It allows data from control (Controls), pre-symptomatic (PreHD) and symptomatic (SympHD) individuals to be compared.

want to compare two objects displayed on the wall by manually placing them next to each another. This is achieved by clicking or tapping on the rectangular bars at the top of a screen, and dragging it to its final destination on the grid. This behaviour has a cascading effect on the data ordering of the grid as depicted in Fig. 9.

### *Interacting with data cubes: the scene and rendering parameters controllers*

Once data is loaded onto one or more screens, the web interface enables synchronous interaction with all displayed data cubes. Figure 10 shows an example interface of the scene controller and the parameters controller for a neuroimaging data cube survey.

The volumes can be rotated, panned, and zoomed interactively (Fig. 10A) using input devices such as a mouse, a track pad or touch pad (e.g., slide, pinch). The interface also provides control over volume rendering properties (e.g., sampling size, opacity and colour map), and adjustment of a single, user-controlled isosurface (Fig. 10B). When rendering parameters are modified (e.g., volume orientation, sampling size, etc.), a JSON string is submitted to the *Manager Unit* and relayed to the *Process-Render Units*, modifying the rendered geometry on the stereoscopic panels of all data cubes.

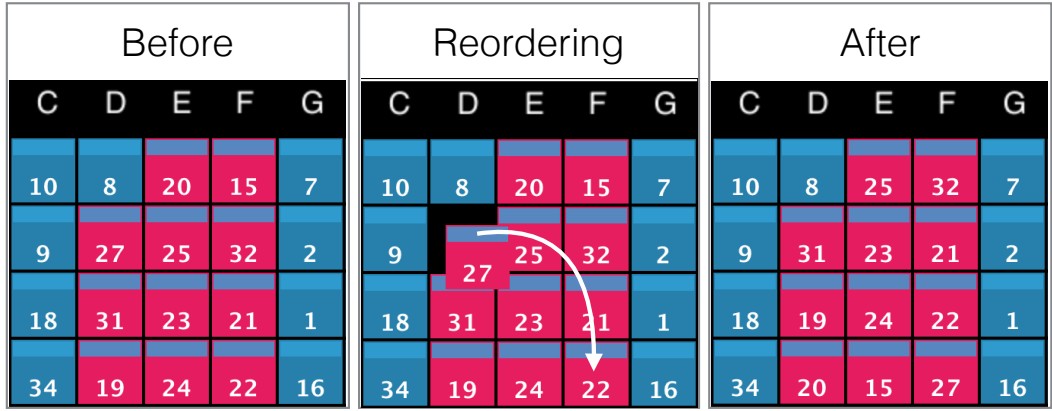

**Figure 9 Example of screens order before, during, and after manual reordering.** The white arrow indicates the clicking and dragging path of the screen (ID: 27) from its original position to its destination. All screens between the origin and the destination will be shifted by one position towards the origin as shown in the panel 'After'.

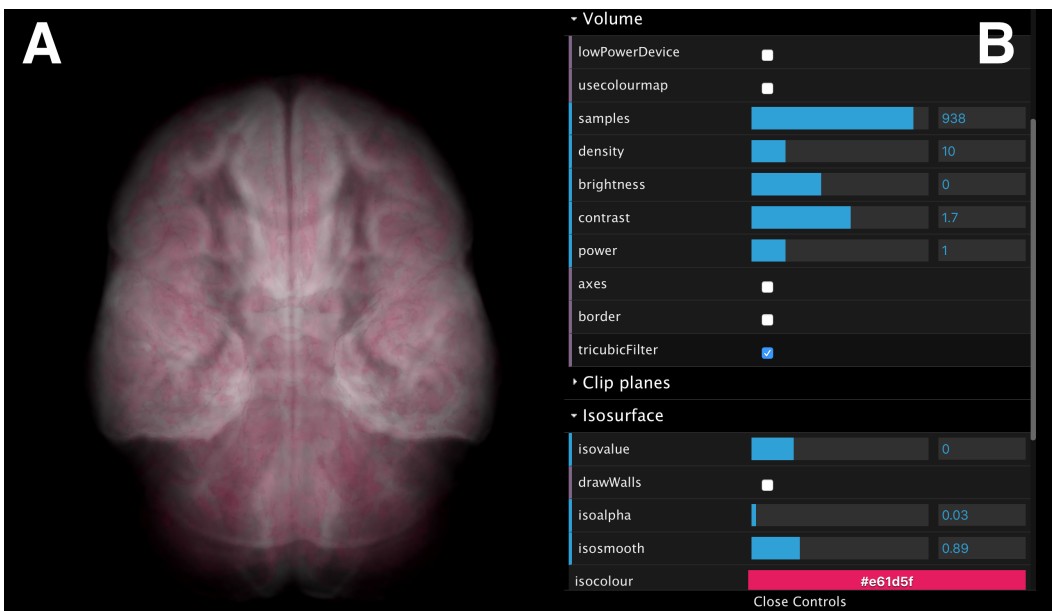

**Figure 10 A screenshot of the interactive web interface.** The interface comprises (A) a scene controller, which displays an interactive reference volume and (B) a set of parameter controls for modifying quantities such as sampling size, opacity, colour map and isosurface level.

### Isosurface and volume rendering in the browser

The interactive controller is based on ShareVol (http://okaluza.github.io/sharevol), and is implemented in WebGL (GLSL language) and JavaScript. It has been developed with an emphasis on simplicity, loading speed and high-quality rendering. The aim being to keep it as a small and manageable library, as opposed to a full featured rendering library such as XTK (https://github.com/xtk) or VolumeRC (https://github.com/VolumeRC). Within the context of the browser and WebGL, we use a one-pass volume ray-casting algorithm,

which makes use of a 2D texture atlas (*Congote et al., 2011*), in order to avoid the use of dynamic server content. By doing so, it minimizes the stress upon the *Manager Unit* by porting the processing onto the client. The texture atlas of volume's slices is pre-processed and provided to the client upon loading. The same shader for both volume rendering and isosurfaces is used at both ends of the system (*Client* and *Process-Render Units*) to provide visualisations as homogeneous as possible.

### An example of execution

A typical encube use-case proceeds as follows. Firstly, the data set is gathered, and a list of survey contents is defined in a semi-structured file (currently CSV format). Secondly, a configuration file is prepared, which specifies information about how and where to launch each *Process-Render Unit*, describes the visualisation system, how to access the data, and so on. Using the *Manager Unit*, each of the *Process-Render Units* is launched.

Once operating, the *Manager Unit* opens TCP sockets to each of the *Process-Render Unit* for further message passing. Still using the *Manager Unit*, the web server is launched to enable connections with *Interaction Units*. Once a *Client* accesses the controller web page, it can start interacting with the displays through the available actions described in 'Interaction Units'.

The workflow serialization, which stores the system's state at each time step, can be reused by piping a specific state to the *Process-Render Units* and/or to the *Interaction Units* to reestablish a previous, specific visualisation state.

An example of executing the application on a local desktop—using a windowed single column of four *Display Units*—is shown in Fig. 11. When using encube locally, it is possible to connect and control the desktop simply by using a mobile device's browser and using a personal proxy server (e.g., Squid (https://help.ubuntu.com/lts/serverguide/squid.html) or SquidMan (http://squidman.net/squidman/index.html)). Using a remote client helps maximizing the usage of display area of the local machine.

## TIMING EXPERIMENT

We systematically recorded loading time and frame rates for both neuroscience data (IMAGE-HD, *Georgiou-Karistianis et al., 2013*), and astronomy data (THINGS, *Walter et al., 2008*). The desktop experiment was completed using Linux (CentOS 6.7), with 16 GB of RAM, 12 Intel Xeon E5-1650 Processors, and a NVIDIA Geforce GTX 470 graphics card. Details about the Monash CAVE2 hardware can be found in 'The CAVE2' and in Table 2. While a direct comparison between both systems is somewhat contrived, we designed our experiment to be representative of a user experience.

*Neuroscience data.* The IMAGE-HD dataset contains tracks files. Tracks represent connections between different brain regions. Each track is composed of a varying number of segments, represented by two spatial coordinates in the 3D space. When loading tracks data, the software calculates a colour for each track—a time consuming task. Pre-processed tracks files—which saves the track color information—can be stored to accelerate the loading process.

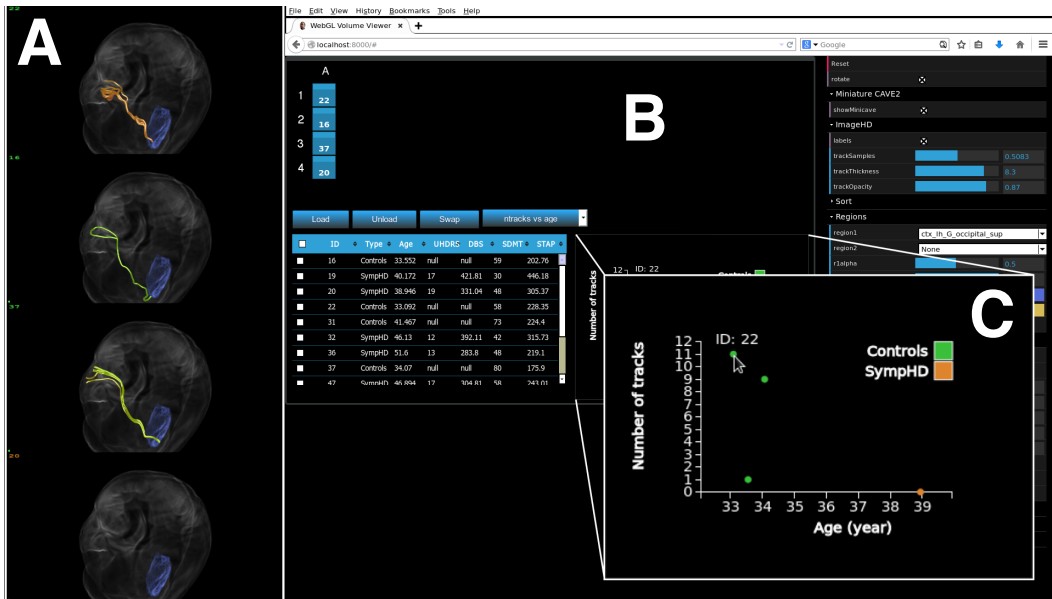

**Figure 11** **encube can be used on a single desktop, hence enabling its use outside the CAVE2 environment.** We show encube running on Linux (CentOS 6.7), with 16 GB of RAM and a NVIDIA Geforce GTX 470 graphics card. The screenshot displays the *Input/output layer*: (A) a *Display Unit* comprising one column of four S2PLOT panels; and (B) an *Interaction Unit*. Four individual brains are plotted (A), along with their internal connections (green and orange tracks) starting from a region of the occipital lobe in the left hemisphere (blue region). A close up of a dynamically generated scatter plot is also shown (C). It gathers quantitative information from all plotted brains, showing the number of displayed tracks per brain in relation with the subject's age. The *Process layer* is running behind the scene.

**Table 3** **Median load time per column (in seconds) using the Monash CAVE2 and a desktop computer for IMAGE-HD data.**

| Measurement | Tracks fraction | Option | CAVE2 | Desktop |
| --- | --- | --- | --- | --- |
| Load time (s) | Subsampled tracks | Unprocessed | 4.82 | 3.48 |
| | | Pre-processed | 0.27 | 0.26 |
| | All tracks | Unprocessed | 5.02 | 20.82 |
| | | Pre-processed | 1.76 | 12.77 |

We report timing for the unprocessed and pre-processed tracks data. In addition, as each file comprises a large number of tracks (e.g., $\geq 2^6$ tracks; $\geq 8 \times 10^7$ points), we present timing when loading all tracks, and when subsampling the total number of tracks by a factor of 40.

The 80 tracks files used in this experiment amounts to ∼39 GB, with a median file size of 493 MB. We repeated the timing experiment three times for each action recorded, and we present the median time in seconds (s). For the Monash CAVE2, we load four files per column, and report the median time from the 20 columns loading data in parallel. On the desktop, we loaded one column of four files. The results are shown in Table 3.

**Table 4** **Median frame rate per column using the Monash CAVE2 and a desktop computer for IMAGE-HD data.** We note that numbers are limited by the refresh rate of the display screens.

| Measurement | Tracks fraction | Render mode | CAVE2 | Desktop |
|---|---|---|---|---|
| S2PLOT area (pixel) | – | – | 1366 × 3072 | 600 × 1200 |
| Frames/s | Subsampled tracks | Mono | 59.5 | 59.6 |
| | | Stereo | 54.5 | – |
| | All tracks | Mono | 4 | 1.5 |
| | | Stereo | 2 | – |

**Table 5** **Median frame rate per column using the Monash CAVE2 and a desktop computer for THINGS data.**

| Measurement | Render mode | CAVE2 | Desktop |
|---|---|---|---|
| S2PLOT area (pixel) | – | 1366 × 3072 | 600 × 1200 |
| Frames/s | Mono | 22.5 | 19.6 |
| | Stereo | 11.2 | – |

To evaluate the rendering speed, in frames per second, we used the auto-spin routine of S2PLOT and recorded the internally-measured S2PLOT frame rate. We report the number of frames per second, along with the S2PLOT area (e.g., Fig. 5). both the number of frames per second, and the number of rendered pixels per second in unit of mega-pixel per second (Mpixel/s), where Mpixel represent $10^6$ pixels. The results are shown in Table 4.

The slight difference in load time may be related to networking, having to fetch data from a remote location, while the desktop simply has to read from a local disk. Nevertheless, timing results are comparable. It is also worth noting that within the CAVE2, after the time reported, 80 volumes are ready for interaction, whereas on the desktop, only four are available.

*Astronomy data.* A similar approach was used for the THINGS dataset. The major difference between the two dataset is the nature of the data. The IMAGE-HD data consist of multiple tracks that needs to be drawn individually using a special geometry shader. A small volume[3] (mean brain, 2 MB) is volume-rendered to help understand the position of the tracks in 3D space relative to the physical brain. In the case of the astronomy data, the data cube is loaded as a 3D texture onto the GPU and volume rendered. In this experiment, we load the entire volume on both the CAVE2 and the desktop.

The 80 data cube files used in this experiment sums up to ≈28 GB, with median file size of 317 MB. In this experiment, the files are not pre-processed. Instead, we load the FITS (http://fits.gsfc.nasa.gov) files directly. On load, values are normalized to the range [0,1] and the histogram of voxels distribution is evaluated. We do not yet proceed with caching. We report the total loading time, which includes normalization and histogram evaluation, and the frame rates obtain with the same methodology as reported in the previous section. Median load time in the CAVE2 and on the desktop is 41.96 and 40.14 s respectively. Frame rate results are in Table 5.

With astronomical data, the loading times per column for CAVE2 and desktop are comparable. that within the CAVE2, after the time reported, 80 volumes are ready for

[3]The volume is stored in the XRW format that can be accessed in parallel. The XRW format is a minimalist volumetric file, stored in compressed binary form. Files contain integer *nx*, *ny* and *nz* pixel dimensions, float *wx*, *wy* and *wz* pixel physical sizes, the data block as unsigned byte per pixel, and a 256 entry RGB colour table using 3 bytes per colour. Tools to read, write and manipulate the XRW format are published at https://github.com/mivp/s2volsurf.

interaction, whereas on the desktop, only four are available. As the number of data cubes (and the number of voxels per cube) increases, the memory will tend to saturate, slowing down the frame rate. However, using the web controller, the user can simply send a camera position which avoids having to rely purely on frame rates and animation to reach a given viewing angle.

As can be expected, the CAVE2 enables much more data to be processed, visualised and analysed simultaneously. For a similar loading period, the CAVE2 provides access to 20 times the data accessible on the desktop. This result comes from the large amount of processing power available, and the distributed model of computation. Nevertheless, the desktop solution can be considered useful in comparison, as researchers can evaluate a subset of data, and rely on the same tools at their office and within the CAVE2.

## DISCUSSION

### About the design

Our system offers several advantages over the classical desktop-based visualisation and analysis methodology where one data cube is examined at a time. The primary advantage is the capability to compare and contrast of order 100 data cubes, each individually beyond the size a typical workstation can handle. Similar to the concept of 'Single Instruction, Multiple Data' (SIMD), this distributed model of processing and rendering leads one requested action to be applied to many data cubes in parallel—which we now call 'Single Instruction, Multiple Views,' and 'Single Instruction, Multiple Queries'. This means that instead of repeating an analysis or a visualisation task over and over from data cube to data cube, the design of encube has the ability to spawn this task to multiple data cubes seamlessly. With this design, it becomes trivial to send a particular request to specific data cubes without (or with minimal) requirements for the user to write code.

As we showed in Fig. 11, the design also permits comparative visualisation and analysis of multiple data cubes on a local desktop—an advantage over the classical, ''one-by-one'' inspection practice. However, this mode of operation has limitations. Depending on the size of the data cubes, only a limited few can effectively be visualised at a time given the amount of RAM the desktop computer contains. Also, the available display area (e.g., on a single 25- inch monitor) is a limiting factor when comparing high-resolution data cubes. When comparing multiple high-resolution data cubes on a single display—depending on the display device—it is likely that they will be compared at a down-scaled resolution due to the lack of display area. Nevertheless, it is feasible, and useful as it enables researchers to use the same tools with or without a tiled-display environment.

It is important to keep in mind that modern computers are rarely isolated; most computers have access to internet resources. In this context, the system design permits the *Process layer* to be located on remote machines. Hence, a researcher with access to remote computing such as a supercomputer or cloud-computing infrastructure could execute the compute-intensive tasks remotely and retrieve the results back to be visualised on a local desktop. In this setting, the communication overhead might become an issue depending on the amount of data required to be transferred over the network. Nevertheless, it may

well be worth it given the processing power gained in the process. For example, it is worth noting that much of the development of encube was accomplished using the MASSIVE Desktop (*Goscinski et al., 2014*). In this mode of operation, it was feasible to load 3 or 4 columns (9 or 12 data sets) on a single MASSIVE node (having typically 192GB RAM).

## About the implementation

As instruments and facilities evolve, they generate data cubes with an increasing number of voxels per cube, which then requires more storage space and more RAM for the data to be processed and visualised. Consider the WHISP and IMAGE-HD projects introduced in 'Background'. Each WHISP data cube varies from ∼32 megabyte to ∼116 megabyte in storage space, and data collected after the update of the Westerbork Synthesis Radio Telescope for the APERTIF survey (*Röttgering et al., 2011*) will be ∼250 GB per data cube (*Punzo et al., 2015*). In the case of IMAGE-HD, the size of a tractography dataset is theoretically unlimited as many tractography algorithms are stochastic. For example, as we zoom into smaller regions of the brain, more tracks can be generated in realtime leading to a larger storage requirement (*Raniga et al., 2012*). To compare a large number of data cubes concurrently, this data volume effectively limits the number of individual file that can be loaded on the desktop computer in its available RAM. Additionally, the display space of desktop is generally limited to one or a few screens.

In the tiled-display context, a clear advantage of our approach is the ability to visualise and compare many data cubes at once in a synchronized manner. *Fujiwara et al. (2011)* showed that comparative visualisation using a large-scale display like the CAVE2 is most effective with sychronised cameras for 3D data. Indeed, we found from direct experimentation in the Monash CAVE2 that synchronized cameras are extremely practical, as they minimize the number of interactions required to manipulate a large number of individual data cubes (up to 80 in our current configuration). In addition, encube enables rendering parameters to be modified or updated synchronously as well.

Another advantage comes through the available display area and available processing power. Not only does the display real estate of the CAVE2 provide many more pixels than a classical desktop monitor, it also enhances visualisation capabilities through a discretized display space. It is interesting to note that bezels around the screens can act as visual guides to organise a large number of individual data cubes. Moreover, our solution can go beyond 80 data cubes at once by further dividing individual screens. This can be achieved in encube by increasing the number of S2PLOT panels from, for example, 4 to 8 or 16 (see Fig. 5).

By nature, the CAVE2 is a collaborative space. Not only can we display many individual data cubes from a survey, but a team can easily work together in the physical space. The physical space gives enough room for researchers to walk around, discuss, debate and divide the workload. The large display space also enables researchers to exploit their natural spatial memory of interesting data and/or similar features. Building on this configuration by adding analysis functionalities via the portable interaction device provides researchers with a way to query and interact with their large dataset while still being able to move around the physical space.

An interesting aspect of the encube CAVE2 implementation is the availability of stereoscopic displays. Historically, most researchers visualise high dimensional data using 2D screens. While 2D slices of a volume are valuable, being able to interact with the 3D data in 3D can lead to an improved understanding of the data, its sub-structures, and so on. Since looking at the data in 3D tends to ameliorate comprehension, we anticipate that the ability to compare multiple volumes *in 3D* should also be beneficial. Moreover, the accessibility to quantitative information enables further understanding about the visualised data. In the current setting, the grid of displayed volumes uses subtle 3D effects that are not tracked to the viewer's position. There is an opportunity to improve this aspect in the future, should it be deemed necessary.

As we mentioned in 'Interaction Units,' further work is required to fully support concurrent users to interact with the system without behavioural anomalies, and to keep multiple devices informed of others' actions. This would represent a major leap towards multi-user data visualisation and analysis of data cube surveys. However, we note that the current system already enables collaborative work where one researcher drives the system (using a *Interaction Unit*), and others researchers observe, query, and discuss the content.

## On the potential to accelerate research in large surveys

Now that we have presented a functional and flexible system, we can speculate on its potential. encube brings solutions to the three common issues of comparative visualisation presented by *Lunzer & Hornbæk* (*2003*, 'Related Work'): (1) requirement of high number of interactions—minimized by the parallelism of action over multiple data cubes; (2) difficulty in remembering what has been previously seen—real-time interactivity and workflow serialization; and (3) difficulty in organising the exploration process—multiple views spread over multiple screens, and multiple mechanisms to organise data and visualisations.

As the system is real-time and dynamic (as shown in 'Timing Experiment'), it represents an additional alternative methodology for researchers to investigate their data. Through interactivity and visual feedback, one does not have time to forget what (s)he was looking for when (s)he triggered an action (load data cube to a panel, query data, rotate the volume). Moreover, by keeping a trail of all actions through workflow serialization, the system acts as a backup memory for the user. A user can look back at a specific environment setup (data organised in a particular way, displayed with a specific angle, with specific rendering characteristics, etc.). Another interesting way of using serialization would be to use checkpointing, where users could bookmark a moment in their workflow, and quickly switch between bookmarks.

Machine learning algorithms, and other related automated approaches, are expected to play a prominent role in classifying and characterize the data from large surveys. As an example of how encube can help accelerate a machine learning workflow, we consider its application to supervised learning.

Supervised learning infers a function to classify data based on a labeled training set. To do so with accuracy, supervised learning requires a large training set (e.g., *Beleites et al., 2013*). The acceleration of data labeling can be achieved via the ability to quickly load, move, and collaboratively evaluate a subsample of a large dataset in a short amount of

time. Extra modules could easily be added to the controller to accomplish such a task. Some form of machine learning could also be incorporated to the workflow in order to characterise data prior to visualisation, and help guide the discovery process. This could lead to interesting research where researchers and computers work collaboratively towards a better understanding of features of interest within the dataset.

Through the use of workflow serialization, users can revisit an earlier session, and more easily return to the CAVE2 and continue where they left off previously. We think that this system feature will help to generate scientific value. Users will not simply get a nice experience by visiting the visualisation space, but they will be supported to gather information for further investigation, that will lead to tangible new knowledge discoveries. Moreover, this feature enables asynchronous collaboration, as collaborators can review previous work from other members of a team and continue the team's steps towards a discovery. We assert that this is a key feature that will help scientists integrate the use of large visualisation systems more easily as part of their work practice.

With the opportunity to look at many data cubes at once at high resolution, it becomes quickly apparent that exploratory science requiring comparative studies will be accelerated. Many different research scenarios can be envisioned through the accessibility of qualitative, comparative and quantitative visualisation of multiple data cubes. For instance, by setting visualisation parameters (e.g., sigma clipping, or a common transfer function), a researcher can compare the resulting visualisations, giving insights about comparable information between data cubes. Similarly, one can compare a single data cube from different camera angles (top-to-bottom, left-to-right, etc.) at a fixed moment in time, or in a time-series (side by side videos seen from a different point-of-view).

Finally, an interesting research scenario of visual discovery using encube would be to implement the lineup protocol from *Buja et al. (2009)*, wherein the visualisation of a single real data set is concealed amongst many decoy (synthetic) visualisations—and trained experts are prompted to confirm a discovery.

## CONCLUSION AND FUTURE WORK

Scientific research projects utilising sets of structured multidimensional images are now ubiquitous. The classical desktop-based visualisation and analysis methodology, where one multidimensional image is examined at a time by a single person, is highly inefficient for large collections of data cubes. It is hard to do comparative work with multiple 3D images, as there are insufficient pixels to view many objects at a reasonable resolution at one time. True collaborative work with a small display area is challenging, as multiple researchers need to gather around the desktop. Furthermore, the available local memory of a desktop limits the number of 3D images that can be visualised simultaneously.

By considering the specific requirements of medical imaging and radio astronomy (as exemplified by the IMAGE-HD project and the WHISP galaxy survey), we identified the CAVE2 as an ideal environment for large-scale collaborative, comparative and quantitative visualisation and analysis. Our solution, implemented as encube, favours an approach that is: flexible and interactive; allows qualitative, quantitative, and comparative visualisation;

provides flexible (meta)data organisation mechanisms to suit scientists; allows the workflow history to be maintained; and is portable to systems other than the Monash CAVE2 (e.g., to a simple desktop computer).

From our review of comparative visualisation systems, we noted that many of the discussed features are desirable for the purpose of designing a system aimed at large-scale data cube surveys. We also noted unexplored avenues. Previous research mainly focused on *rendering a visualisation seamlessly over multiple displays* and *enabling interaction with the visualisation space*. However, the following questions remained open: (1) *how to integrate comparative visualisation and analysis into a unified system*; (2) *how to document the discovery process*; and (3) *how to enable scientists to continue the research process once back at their desktop*.

Building on these desirable feature, we presented the design of encube, a visualisation and analysis system. We also presented an implementation of encube, using the CAVE2 at Monash University as a testbed. We also showed that encube can be used on a local machine, enabling the research started in the CAVE2 to be continued at one's office, using the same tools. The implementation is a work in progress. More features and interaction models can, and will, be added as researchers request them. We continue to develop encube in collaboration with researchers, so that they can integrate our methods into their work practices.

## ACKNOWLEDGEMENTS

This work was enabled and supported by the Monash Immersive Visualisation Platform (http://monash.edu/mivp).

### Funding

The IMAGE-HD study was funded by the CHDI Foundation, Inc (USA). This research was supported by the VLSCI's Life Sciences Computation Centre, an initiative of the Victorian Government, Australia hosted at the University of Melbourne. The Fonds de recherche du Québec—Nature et technologies (FRQNT) and Swinburne Research for postgraduate scholarships provided support to DV. The Victorian Life Sciences Computation Initiative (VLSCI) provided support for a summer student scholarship to YB which supported parts of this work. The funders had no role in study design, data collection and analysis, decision to publish, or preparation of the manuscript.

### Grant Disclosures

The following grant information was disclosed by the authors:
CHDI Foundation, Inc (USA).
VLSCI's Life Sciences Computation Centre.
Fonds de recherche du Québec—Nature et technologies (FRQNT).
Swinburne Research.
Victorian Life Sciences Computation Initiative (VLSCI).

## Competing Interests

The authors declare there are no competing interests.

## Author Contributions

- Dany Vohl conceived and designed the experiments, performed the experiments, analyzed the data, contributed reagents/materials/analysis tools, wrote the paper, prepared figures and/or tables, performed the computation work, reviewed drafts of the paper.
- David G. Barnes conceived and designed the experiments, performed the experiments, analyzed the data, contributed reagents/materials/analysis tools, wrote the paper, performed the computation work, reviewed drafts of the paper.
- Christopher J. Fluke conceived and designed the experiments, performed the experiments, contributed reagents/materials/analysis tools, wrote the paper, performed the computation work, reviewed drafts of the paper.
- Govinda Poudel conceived and designed the experiments, reviewed drafts of the paper.
- Nellie Georgiou-Karistianis conceived and designed the experiments, contributed reagents/materials/analysis tools, reviewed drafts of the paper.
- Amr H. Hassan conceived and designed the experiments, performed the experiments, wrote the paper, performed the computation work, reviewed drafts of the paper.
- Yuri Benovitski performed the experiments, contributed reagents/materials/analysis tools, performed the computation work.
- Tsz Ho Wong contributed reagents/materials/analysis tools, performed the computation work.
- Owen L. Kaluza and Toan D. Nguyen conceived and designed the experiments, performed the experiments, analyzed the data, contributed reagents/materials/analysis tools, performed the computation work.
- C. Paul Bonnington contributed reagents/materials/analysis tools.

## Data Availability

Raw data: https://github.com/mivp/encube.

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
