# Peer review of "Large-scale comparative visualisation of sets of multidimensional data"

_PeerJ Computer Science, doi:10.7717/peerj-cs.88_

## Round 0.1 · original submission · Major Revisions

The technical quality of the draft could be further improved. The scientific contributions should be clearly stated and supported by data thus the reader can perceive the novelty of this work.

·

Basic reporting

The article is clear and has a correct structure. Motivation is duly substantiated and related work is extensive and updated.

Experimental design

Section 3 is somewhat generic in terms of the design description. Section 4, and more specifically 4.3, provides much space to the description of the control software implementation, but only describes a few examples.
It could be explained in this section the appropriate set of tools and functionalities for the analysis of the two proposed problems: Magnetic Resonance Imaging (MRI) data from IMAGE-HD and large-scale systematic morphological classification of the kinematic structures of galaxies.
Regard this second problem is cited in the article but later nothing is explained about him.

Validity of the findings

The developed system, ENCUBE, has some excellent features such as:
It allows comparative visualization and analysis of large amounts of data, actions aplied to data cubes in parallel, large display visualization area, very high processing power, collaborative workspace, stereoscopic capability and workflow serialization.

Additional comments

It is a good tool for scientific visualization.

Reviewer 2 ·

Basic reporting

No Comments.

Experimental design

The submission must describe original primary research within the Scope of the journal. -> Ok
The submission should clearly define the research question, which must be relevant and meaningful. The knowledge gap being investigated should be identified, and statements should be made as to how the study contributes to filling that gap. -> There is no research question.
The investigation must have been conducted rigorously and to a high technical standard. -> There is no experiment.
Methods should be described with sufficient information to be reproducible by another investigator. -> Ok
The research must have been conducted in conformity with the prevailing ethical standards in the field. -> Ok

Validity of the findings

The data should be robust, statistically sound, and controlled. -> There is no data gathering.
The data on which the conclusions are based must be provided or made available in an acceptable discipline-specific repository. -> All conclusions are speculative.
The conclusions should be appropriately stated, should be connected to the original question investigated, and should be limited to those supported by the results. -> This is not ok in the writing.
Speculation is welcomed, but should be identified as such. -> It is not identified as speculative.
Decisions are not made based on any subjective determination of impact, degree of advance, novelty, being of interest to only a niche audience, etc. Replication experiments are encouraged (provided the rationale for the replication, and how it adds value to the literature, is clearly described); however, we do not allow the ‘pointless’ repetition of well known, widely accepted results. -> The conclusions are mostly subjective.
Negative / inconclusive results are acceptable. -> There are no experiments.

Additional comments

The article describes encube, a data visualization deviced aimed at the exploration of large data.

There is no doubt that the authors have put great effort into this work, and that it seems to present relevant features. However, the text has some serious flaws. Most of them seem to be linked to the excessive description of the system, which makes the paper miss points that are relevant in scientific literature: evaluation and the detection of the contributions of the work using objective data.

These are my comments:

“For a comprehensive review of the variety of standard techniques see, for
example, Akenine-M ̈oller et al. (2008), Toriwaki and Yoshida (2009) and Szeliski (2010).”
> It would be very useful for the reader to have some (two? Three?) figures with examples of standard techniques and how they work. Also, why are there three review papers on the subject in three consecutive years? It can be the case that there was a lot of innovation in the field in these years. Could the authors state what is the contribution of each of these reviews?

“Structured three dimensional (3D) images or data cubes are ubiquitous in scientific research.”
> I disagree with that. There are many fields in science that are oblivious about data cubes. Again, some figures and a deeper discussion would greatly increase the appeal of this paper.

> Along “Related Work”: again, I miss some figures that could highlight the differences between visualization systems proposals and the preceeding ones. It would probably be a good idea to only select the two or three that provided inspiration for encube.

In Section 5.3
> Without an user study, all of these discussions are based on anedoctal data or speculation. This article needs an user study. My next note also regards this question.

> Although the authors state that there are three important questions (“1) how to integrate comparative visualisation and analysis into a unified system; 2) how to document the discovery process; and 3) how to enable scientists to continue the research process once back at their desktop.”). However:
> 1) I am not convinced that this was discussed along the article. Again, the reader needs at least some comparative figures displaying how encube is different from previous systems. Also, this only accounts for the proposal of the system: proving that this point was properly addressed requires an user study.
> 2) Again, an user study is necessary to validate the proposed system. How did users interact with the discovery process history?
> 3) By research, I am guessing the authors mean “discovery” or “exploration”? The server-based service is a good solution for this. The user study, however, should highlight whether the server-based approach actually improved the data exploration experience.

I believe that the core of the work (the proposed system) is of good quality, and that the authors should calmly address the issues above prior to publishing.

---

## Round 0.2 · accepted · Accept

The authors have made substantial changes to the initial draft. In particular, a new timing experiment has been conducted for neuroscience and astronomical datasets, where CAVE2 is compared to a personal desktop computer. There are also minor changes in section 5.3, and also a compared analysis with other systems, along with many amendments and improvements of the explanation.

Not all the suggestions of rev 2 have been followed. In particular, figures for standard techniques have not been included, however the justification given by the authors is admissible. Generally speaking, explanations in the rebuttal letter are reasonable.

---

## Author Rebuttal · Round 0.2

**Centre for Astrophysics & Supercomputing**
Swinburne University of Technology
1 Alfred Street
Hawthorn VIC Australia 3122
Tel +61 3 9214 8708
dvohl@swin.edu.au

[Figure]

August 24th, 2016

Dear Professor Sánchez,

Vohl *et al.*, "Large-scale comparative visualisation of sets of multidimensional data", ref: #CS-2016:05:10577:0:1:REVIEW, Article ID: 10577.

We thank the reviewers for their time and generous comments on the manuscript, which we have revised to address their concerns.

We are pleased to find both reviewers acknowledged our work as being of good quality. Following their detailed and helpful suggestions, we have refined our manuscript to better highlight our research questions, our scientific contribution, and how they relate to literature. We also included a new experiment section, along with additional tables, figures, and examples highlighting how *encube* meets our user requirements for our two targeted test cases (medical imaging and astronomy), where other systems do not. We also highlight that part of the discussion speculates on *encube*'s potential based on the system's features presented throughout the article.

We gave a great deal of consideration to Reviewer 2's request for a user study – noting that Reviewer 1 did not identify that such a study was required. The primary focus of this paper is to present our technical solution solving our requirements for visualization led discovery in large scale data cube surveys; requirements previously unmatched based on our review of previous work. We believe a user study would change the direction and purpose of the work we are presenting in this paper. Hence, we do not consider a user experiment for this particular paper.

We believe concepts we introduce, namely integrating comparative and quantitative visualization and analysis into a unified system, and workflow serialization in the context of tiled-displays are novel.

We believe the article is now suitable for publication in PeerJ Computer Science.

Dany Vohl
Ph.D. Candidate
On behalf of all authors.

We respond here to the specific comments and feedback of the Editor and Reviewers.

*Editor (Luciano Sánchez)*
*The technical quality of the draft could be further improved. The scientific contributions should be clearly stated and supported by data thus the reader can perceive the novelty of this work.*

**We have modified our manuscript to improve the technical quality of the draft. We also modified portions of the article, including the introduction, to clearly state our scientific contributions, which is supported by data (we have a functional piece of software). We introduced a new section, Section 5, presenting the results of a timing experiment comparing performances of *encube* when used within the CAVE2 and on a personal desktop computer. The article also presents real world results, described in the text and figures.**

*Reviewer 1 (Luis Junco)*
*"Section 3 is somewhat generic in terms of the design description."*
**We have modified the section header from "Design" to "Overview of *encube",* clarifying that we are outlining the different parts and concepts of encube that will be further explained in Section 4.**

*"Section 4, and more specifically 4.3, provides much space to the description of the control software implementation, but only describes a few examples. (...)"*
**We have modified Section 4.3.2 (in particular "Integrating comparative visualisation and analysis as a unified system") to better highlight our two functionalities used as proof of concepts, both for astronomy (interactive histogram) and medical imaging (automatically generated scatter plot, showing the number of neural connections for all visualised brains).**

*"It could be explained in this section the appropriate set of tools and functionalities for the analysis of the two proposed problems (...) Regard this second problem is cited in the article but later nothing is explained about him."*
**We do not discuss all potential features that scientists in both field may want to implement as it would be outside the scope of the paper. We do mention however that additional functionalities are to be implemented based on specific user and use-case requirements. In addition, we added an extra panel to Figure 7 displaying several galaxy morphologies visualised with *encube* to give the reader an idea of what galaxy morphologies can look like.**

*Reviewer 2*
*"There is no research question."*
**As pointed out later on by Reviewer 2, our research questions are related to : 1) how to integrate comparative visualisation and analysis into a unified system; 2) how to document the discovery process; and 3) how to enable scientists to continue the research process once back at their desktop.**

We modified the introduction from an active form (e.g. We extend these solutions by considering [these three points]) to question form to better accentuate our research questions.

*"There is no experiment."; "There is no data gathering."*
As shown in figures 7 and 10, we have experimented with our system to show that: 1) we properly enabled both comparative visualisation and quantitative analysis to run dynamically; 2) the system properly enables integration of real-world functionalities such as the interactive histogram for astronomy, the interactive scatterplot for medical imaging, and the recording of workflow; 3) the system can be executed as a standalone software.

In addition, we conducted a new timing experiment where we evaluate loading time and frame rate for both neuroscience and astronomical datasets. This experiment highlights the responsiveness of the system while dealing with multiple large files simultaneously. Results of this experiment are now included as Section 5. In this section, we compare performances (load time and frame rate) between *encube* running within the Monash CAVE2 and on a personal desktop computer.

We report that for a similar loading period, the CAVE2 provides access to 20 times the data accessible on the desktop. We also report that frame rates vary depending on the rendering method (e.g. tracks versus volume rendering; rendering in mono or in stereo).

*"The conclusions should be appropriately stated, should be connected to the original question investigated, and should be limited to those supported by the results. -> This is not ok in the writing."*
We modified our manuscript to better connect our conclusions with our original questions.

*"Speculation is welcomed, but should be identified as such. -> It is not identified as speculative.";*
*"The conclusions are mostly subjective."*
We modified the title for section 5.3 and clarified its discussion to better highlight that it includes speculations based on the real system's features described throughout the article.

*"There is no doubt that the authors have put great effort into this work, and that it seems to present relevant features. However, the text has some serious flaws. Most of them seem to be linked to the excessive description of the system, which makes the paper miss points that are relevant in scientific literature: evaluation and the detection of the contributions of the work using objective data."*

We included a new table (Table 1) comparing our system with previous work based our user requirements. We show that none of the previous systems met all requirements. In particular, none of the previous systems included quantitative mechanisms to evaluate visualised data. We also include new interaction mechanisms for comparative visualisation (see Table 1). Furthermore, none of the previous work considered recording the workflow history – which is a key feature to continue the data exploration subsequently to a single session. Finally, none of the previous work discussed using their software as a standalone solution. Since not all

researchers have continuous access to a CAVE2 or other tiled-display, this new feature provides a means to continue the comparative work outside of an advanced display environment.

*"It would be very useful for the reader to have some (two? Three?) figures with examples of standard techniques and how they work. Also, why are there three review papers on the subject in three consecutive years? It can be the case that there was a lot of innovation in the field in these years. Could the authors state what is the contribution of each of these reviews?"*

**We modified the first paragraph of the introduction to highlight the differences between all three references. We did not include figures for standard techniques as this is outside the scope of this paper.**

*"I disagree with that. There are many fields in science that are oblivious about data cubes. Again, some figures and a deeper discussion would greatly increase the appeal of this paper."*

**We modified the statement and included a paragraph giving examples of fields that use data cubes in their science.**

*"Along "Related Work": again, I miss some figures that could highlight the differences between visualization systems proposals and the preceeding ones. It would probably be a good idea to only select the two or three that provided inspiration for encube."*

**We added a figure in Section 2 showing system design schematics from the four systems that provided inspiration for *encube*. In addition, in Section 3, we added a table (Table 1) comparing these systems with ours regarding our user requirements.**

*"In Section 5.3*
*> Without an user study, all of these discussions are based on anedoctal data or speculation."*

**We modified the title for section 5.3 and clarified its discussion to better highlight that it includes speculations based on the real system's features described throughout the article.**

*"Although the authors state that there are three important questions ("1) how to integrate comparative visualisation and analysis into a unified system; 2) how to document the discovery process; and 3) how to enable scientists to continue the research process once back at their desktop."). However (...)*

*(…) "1) I am not convinced that this was discussed along the article. Again, the reader needs at least some comparative figures displaying how encube is different from previous systems. Also, this only accounts for the proposal of the system: proving that this point was properly addressed requires an user study."*

**We added a mention in the caption of Figure 2 to better highlight this integration. We also renamed a subsection, now called "Integrating comparative visualisation and analysis as a unified system", in Section 4.3.2 to better highlight this point. In a sense, the article describes in detail how this integration is made possible.**

**We gave a great deal of consideration to the request for a user study. This work commenced because of our long-term collaboration with researchers from the IMAGE-HD study (including**

co-authors Georgiou-Karistianis, Poudel and Barnes) and with astronomers working on galaxy survey projects (including co-author Fluke). These research teams are facing specific challenges that we are addressing. The alternative solutions that we reviewed did not have the full set of capabilities that these two research teams needed. Current expertise and experience with desktop-bound solutions has shown us that they are not adequate to the type of large-scale classification tasks that motivated our requirements (section 1.1). *Encube* has developed to a stage that we are now bringing the science team leaders, and associates, into the CAVE2 to make progress on their specific research goals. Future publications will focus on the scientific outcomes and advances in early identification of Huntington's disease and morphological (shaped-based) analysis of neutral hydrogen in galaxies.

The primary focus of this paper is to present and make available a technical solution that satisfies new requirements in the "big data" era, hitherto not delivered by existing visualisation solutions. The technical solution needs to be fit for purpose, which it is. In practical terms, the technical solution and its tools may then be used by research teams in their own workflow process. We believe it is this workflow that must be validated as a whole by a user study, not the individual tools. Hence, we do not consider a user experiment for this particular paper. We do however consider a timing experiment, which is now included as Section 5.

*(…) "2) Again, an user study is necessary to validate the proposed system. How did users interact with the discovery process history?"*
We did provide a description of mechanisms to store the workflow, which shows how such a concept can be achieved. Testing how users interact with it is another research question that we do not tackle in this study.

*(…) "3) By research, I am guessing the authors mean "discovery" or "exploration"? The server-based service is a good solution for this. The user study, however, should highlight whether the server-based approach actually improved the data exploration experience."*
We did show that our system can be work on a single machine, and still execute comparative and quantitative visualisations (see Figure 10 and Section 5). Testing how users continued their data exploration is another research question that we do not tackle in this study.